# Cefto Real-Life Study: Real-World Data on the Use of Ceftobiprole in a Multicenter Spanish Cohort

**DOI:** 10.3390/antibiotics12071218

**Published:** 2023-07-21

**Authors:** Carmen Hidalgo-Tenorio, Inés Pitto-Robles, Daniel Arnés García, F. Javier Membrillo de Novales, Laura Morata, Raul Mendez, Olga Bravo de Pablo, Vicente Abril López de Medrano, Miguel Salavert Lleti, Pilar Vizcarra, Jaime Lora-Tamayo, Ana Arnáiz García, Leonor Moreno Núñez, Mar Masiá, Maria Pilar Ruiz Seco, Svetlana Sadyrbaeva-Dolgova

**Affiliations:** 1Unit of Infectious Diseases, Hospital Universitario Virgen de las Nieves, Instituto de Investigación Biosanitario de Granada (IBS-Granada), 18012 Granada, Spain; ipittorobles@gmail.com (I.P.-R.); arnesgarciadaniel@gmail.com (D.A.G.); 2CBRN and Infectious Diseases, Hospital General Defensa Gómez Ulla, 28047 Madrid, Spain; javimembrillo@gmail.com; 3Infectious Diseases Service, Hospital Clinic, 08036 Barcelona, Spain; lmorata@clinic.cat; 4Pneumology Deparment, Hospital Universitario La Fe, Valencia (CIBERES), 46026 Valencia, Spain; rmendezalcoy@gmail.com; 5Internal Medicine Service, Hospital La Moraleja, 28050 Madrid, Spain; olgabravodepablo@gmail.com; 6Infectious Diseases Service, Hospital General of Valencia, 46014 Valencia, Spain; vicente.abril.lopezdemedrano@gmail.com; 7Infectious Diseases Service, Hospital Universitario La Fe, Valencia (CIBERES), 46026 Valencia, Spain; salavert_mig@gva.es; 8Infectious Diseases Service, Hospital Ramón y Cajal, 28034 Madrid, Spain; pilar1vizcarra@gmail.com; 9Internal Medicine Service, Hospital Universitario 12 Octubre (CIBERINFEC), 28041 Madrid, Spain; sirsilverdelea@yahoo.com; 10Department of Infectious Diseases, Hospital Sierrallana, 39300 Torrelavega, Spain; anam.arnaiz@scsalud.es; 11Internal Medicine Service, Hospital Fundación de Alcorcón, 28922 Alcorcón, Spain; lmorenon@salud.madrid.org; 12Infectious Diseases Service, Hospital Universitario General of Elche, 03203 Elche, Spain; marmasiac@gmail.com; 13Internal Medicine Service, Hospital Infanta Sofía, 28702 Madrid, Spain; mprseco@salud.madrid.org; 14Pharmacy Service, Hospital Universitario Virgen de las Nieves, Instituto de Investigación Biosanitario de Granada (IBS-Granada), 18012 Granada, Spain; sadyrbaeva@gmail.com

**Keywords:** ceftobiprole, sepsis, older, real-world data, OPAT

## Abstract

Background: Ceftobiprole is a fifth-generation cephalosporin that has been approved in Europe solely for the treatment of community-acquired and nosocomial pneumonia. The objective was to analyze the use of ceftobiprole medocaril (Cefto-M) in Spanish clinical practice in patients with infections in hospital or outpatient parenteral antimicrobial therapy (OPAT). Methods: This retrospective, observational, multicenter study included patients treated from 1 September 2021 to 31 December 2022. Results: A total of 249 individuals were enrolled, aged 66.6 ± 15.4 years, of whom 59.4% were male with a Charlson index of four (IQR 2–6), 13.7% had COVID-19, and 4.8% were in an intensive care unit (ICU). The most frequent type of infection was respiratory (55.8%), followed by skin and soft tissue infection (21.7%). Cefto-M was administered to 67.9% of the patients as an empirical treatment, in which was administered as monotherapy for 7 days (5–10) in 53.8% of cases. The infection-related mortality was 11.2%. The highest mortality rates were identified for ventilator-associated pneumonia (40%) and infections due to methicillin-resistant *Staphylococus aureus* (20.8%) and *Pseudomonas aeruginosa* (16.1%). The mortality-related factors were age (OR: 1.1, 95%CI (1.04–1.16)), ICU admission (OR: 42.02, 95%CI (4.49–393.4)), and sepsis/septic shock (OR: 2.94, 95%CI (1.01–8.54)). Conclusions: In real life, Cefto-M is a safe antibiotic, comprising only half of prescriptions for respiratory infections, that is mainly administered as rescue therapy in pluripathological patients with severe infectious diseases.

## 1. Introduction

There has been a disturbing increase in multi-resistant microorganisms worldwide over the past decade [1], presenting clinicians with major diagnostic and therapeutic challenges. This phenomenon has been associated with a rise in the failure of empirical antibiotic therapies [2] and with a delay before the administration of an effective drug [3], thereby increasing mortality rates [4]. The rate of carbapenemase-resistant *Pseudomonas* spp. is currently >20% in Spain [1], mainly due to efflux pumps and porin losses. Therefore, carbapenem sparing strategies are recommended to attempt to decrease the rate of carbapenemase-producing Enterobacteriaceae. A randomized controlled trial (MERINO) reported a lower mortality rate using meropenem than using piperacillin/tazobactam in patients with ceftriaxone-resistant *Escherichia coli* or *Klebsiella pneumoniae* bloodstream infections. The findings did not support the utilization of piperacillin-tazobactam against these infections [5]. This has fostered the administration of bactericide antibiotics other than piperacillin/tazobactam to treat gram-negative bacteria such as *P. aeruginosa*, including ceftobiprole. Ceftobiprole medocaril (Cefto-M) is a broad-spectrum, fifth-generation cephalosporin against gram-negative cocci and bacilli, ranging from methicillin-resistant *S. aureus* (MRSA) to ampicillin-susceptible *Enterococcus faecalis*, *faecium,* and *P. aeruginosa*. It is not affected by efflux pumps or porin losses [6]. It has a spectrum of potential interest for the treatment of catheter-related bacteremia, endocarditis, or complicated urine infections. In an experimental study, the bactericide capacity of Cefto-M in biofilm was higher than that of linezolid, vancomycin, or daptomycin against infections caused by MRSA, methicillin-susceptible *S. aureus* (MSSA), or coagulase-negative *staphylococci* (CoNS) [7]. It may, therefore, be useful for treating infections related to devices (intracardiac, cranial leads, etc.), prosthetic valves, endoprostheses, or osteosynthesis materials. It has demonstrated a similar effectiveness to that of other antibiotics in skin and soft tissue infections [8]. Nevertheless, it has only been approved in Europe for the treatment of community-acquired (CAP) and nosocomial (NP) pneumoniae, excluding ventilator-associated pneumonia (VAP).

Clinical trials are the gold standard for approving novel pharmaceutical products or therapies. However, they can differ from actual clinical experience due to their strict eligibility criteria and optimal conditions. Real-world data can help bridge this gap, thereby supporting and accelerating the incorporation of effective new therapies and technologies into routine clinical practices [9]. However, sample sizes have been limited in previous real-life studies on Cefto-M [10]. With this background, this real-life study in Spain was designed to examine the routine administration of Cefto-M in patients with any type of infection in hospital or receiving outpatient parenteral antimicrobial therapy (OPAT), considering the health and safety outcomes and the mortality-related factors.

## 2. Results

### 2.1. Cohort Description

The study included 249 individuals with a mean age of 66.6 ± 15.4 years. A total of 59.4% were male and 92.8% were Caucasian with a mean age-adjusted Charlson index of four (IQR 2–6) and 49.4% had cardiovascular risk factors, primarily cardiovascular disease (31.3%), arterial hypertension (29.3%), and diabetes mellitus (28.1%). A total of 20.9% were immunosuppressed, 14.1% had chronic kidney failure, and 11.6% had chronic obstructive pulmonary disease (COPD) (Table 1). The infection origin was nosocomial/healthcare-related in 57% of the patients. Cefto-M was administered in hospital to 95.6% of the patients (80.4% in the medical department) and as OPAT in 4.4% of the patients. Sepsis was present in 26.5%, septic shock in 4.4%, and concomitant COVID-19 infection in 13.7% of the patients. The median number of foci was one (IQR: 1–1). The type of infection was respiratory in 55.8% (CAP in 24.1%, NP in 24.9%, and VAP in 2%); skin and soft tissue infection (SSTI) in 21.7%; and bacteremia in 17.7% of the patients (catheter-related in 2.8% and no focus in 14.9%) (Table 1).

### 2.2. Microbiological Isolation

Microbiological isolates were obtained from 137 patients (55%) and were polymicrobial in 56 (40.6%). Among the isolates, 87 (35.3%) were gram-positive cocci (GPC), 20 (22.9%) of which were coagulase-negative staphylococci (CoNS), including 13 (65%) that were methicillin-resistant. A total of 46 (18.4%) were *S. aureus*, including 21 (45.6%) methicillin-susceptible *S. aureus* (MSSA) and 24 (52.3%) methicillin-resistant *S. aureus* (MRSA) isolates. A total of nine (10.3%) were *Enterococcus* spp., including eight (88.9%) *E. faecalis* and one (11.1%) ampicillin-susceptible *E. faecium* isolates. A total of 10 (11.5%) were *Streptococcus* spp., including five (50%) *S. pneumoniae* and five (50%) *Streptococci* of other species. A total of 49 were gram-negative bacilli (GNB), including 13 (26.5%) multi-susceptible *Enterobacteriaceae,* 31 (63.3%) non-fermenting GNB (100% *P. aeruginosa*), and five (10.2%) GNB of other species (*Hemophilus influenzae* [2], *Morganella* spp. [2], and *Moraxella* spp. [1]). Table 2 lists the other variables.

All the isolated microorganisms treated with Cefto-M were susceptible to this drug (three MRSA, three MSSA, one enterococcus, one streptococcus, and 10 GNB, including four *P. aeruginosa*). Among the GPC, 97.2% (*n* = 35) were susceptible to vancomycin (100% of MRSA, 93.3% of MSSA, and 100% of both enterococci and streptococci). In terms of the GNB susceptibility, 83.3% of the *P. aeruginosa* isolates were susceptible to meropenem, 40% to cefepime, and 70% to piperacillin/tazobactam (Table 3).

### 2.3. Outcomes

The median (IQR) stay was 20 (13–32) days. The total Cefto-M dose per patient was 10.5 (7.5–15) g for 7 days (5–10), what was administered in monotherapy to 134 patients (53.8%). It was prescribed as an empirical antibiotic treatment in 67.9% of the patients, and was appropriate in 82.8% of these. It was used as a first-line antibiotic in 74 (29.7%) patients and a second-line or more in 176 (70.3%). It was administered due to the failure of previous antibiotic therapy in 33.7% of the patients and after receiving the microbiology results from 26.1%. The death of 54 patients (21.7%) during the 6-month follow-up was directly attributable to infection in 28 (11.2%) patients, 17 (60.7%) of whom died during the first 14 days, nine (32.1%) between days 15 and 28, and two (7.1%) between day 29 and 6 months. Readmission for the same reason was recorded in 15 patients (6%) and for recurrence during the first month of follow-up in three (1.2%) (Table 4).

The mortality rate by infection type was 16.7% (10/60) for CAP, 14.5% (9/62) for NP, 40% (2/5) for VAP, 11.4% (5/44) for bacteremia, 5.6% (3/54) for SSTI, and 20% (7/34) for concomitant COVID-19 infection (Figure 1).

The mortality rate was 9.1% (8/88) for infections caused by GPC (MRSA 20.8% [5/24], *E. faecalis* 12.5% [1/8], MSSA 9.5% [2/21], CNS-MR 0% [0/13], *Pneumococcus* 0% [0/5], *E. faecium* S-ampicillin 0% [0/1], *S. pneumoniae* 0% [0/5], and *Streptococcus* spp. 0% [0/5]). The mortality rate was 11.8% (6/51) for infections caused by GNB (*P. aeruginosa* 16.1% [5/31], multi-susceptibility *Enterobacteriaceae* 0% [0/12], and other non-fermenting GNB 0% [0/2]), and 0% in infections by gram-positive bacilli (0/1) (Figure 2).

### 2.4. Adverse Effects

No adverse effect was recorded in 96.4% of the treated patients, a mild effect in 1.6%, and a moderate effect in 1.6%. No patient abandoned the treatment due to adverse effects. Mild hypertransaminasemia was reported in 1.2% of the patients; diarrhea, nausea, and vomiting in 0.8%; and skin rash in 0.4% (Table 5).

### 2.5. Bi- and Multivariate Analyses of Mortality-Related Factors

In the bivariate analysis, mortality was associated with higher age (76.7 ± 13.3 vs. 65.3 ± 15.2 yrs.; *p* = 0.0001), ICU admission (28.6 vs. 2.1%; *p* = 0.001), cardiovascular risk factors (78.6 vs. 45.7%, *p* =0.001), underlying neurological disease (21.4 vs. 6.8%; *p* = 0.019), immunodepression (35.7 vs. 19%; *p* = 0.04), sepsis/septic shock (57.1 vs. 27.6%; *p* = 0.0001), VAP (7.1 vs. 1.4%, *p* = 0.04), fewer days of Cefto-M treatment (six [P25–P75: 3–8.5] vs. seven [P25–P75: 5–10] days, *p* = 0.029), and a lower total dose (in mg) of Cefto-M (nine [4.5–12.75] vs. 10.5 [7.5–15], *p* = 0.049). Hospitalization in a department/unit of infectious diseases emerged as a protective factor (24.9% vs. 7.1%; *p* = 0.035).

In the multivariate analysis, the factors associated with infection-related mortality were age (OR: 1.1 95% CI [1.04–1.16]), sepsis/septic shock (OR 2.94, 95% CI [1.01–8.54]), and ICU admission (OR 42.02, 95% CI [4.49–393.4]) (Table 6).

## 3. Discussion

The patients in this real-life study were elderly, largely male, and pluripathological, with a high comorbidity index and a predominance of cardiovascular risk factors. Around one in five were immunodepressed, one in seven had kidney failure, and one in ten had COPD. More than half of the infections were nosocomial or healthcare-related, and approx. 5% received OPAT. As in the case of other beta-lactams, the pharmacokinetics and pharmacodynamics of Cefto-M favor its infusion for 24 h, making it a potentially useful antibiotic for OPAT regimens in the patients with infections caused by GPC, including MRSA and ampicillin-susceptible *Enterococcus* spp., and by non-ESLB-producing GNB such as *Pseudomonas* spp. [11].

More than one-third of the participants had sepsis/septic shock, and one-seventh were co-infected with SARS-CoV-2 (COVID-19). Septic shock was described as an independent mortality risk factor with an increase in the risk of up to 12% for every hour in shock, regardless of the focus, isolate, type of poly/monomicrobial infection, or presence/absence of bacteremia [12]. A multicenter study of more than 5000 individuals with septic shock reported a mortality rate of approx. 50% when the antibiotic treatment was appropriate and 89% when it was not [13]. Co-infection with SARS-CoV-2 in critical patients with NP or VAP has been known to worsen the prognosis, although it does not increase the rate of invasive fungal infection or change the type of microorganism isolated at respiratory level [14]. In the present study, only approx. half of the patients received Cefto-M for respiratory infections (half NP and half CAP), which is the sole indication for this antibiotic in Spain [15]. One-fifth of the patients were treated for skin/soft tissue infections and one-sixth for bacteremia. Cefto-M was effective against *Enterococcus* in a murine model of a UTI [16] and was proposed as a possible treatment for a complicated UTI produced by *Pseudomonas* spp. [17]. Three non-inferiority clinical trials in the patients with skin and soft tissue infections reported no difference between Cefto-M and its comparators in terms of clinical or microbiological responses or safety profiles [18]. The decisions of clinicians to prescribe Cefto-M to the remaining patients in this real-life study were supported by pharmacokinetic [19] and in vitro [20] studies. In addition, Cefto-M was used to treat gram-negative bacterial (GNB) infections to avoid the utilization of carbapenems and help reduce the incidence of carbapenemase-producing *Enterobacteriaceae.* Furthermore, in the cases of infection caused by methicillin-resistant CGP such as MRSA, which were all susceptible to vancomycin, Cefto-M was prescribed instead of this lipoglycopeptide due to its rapid bactericidal activity, high volume of distribution to tissues, and excellent safety profile. Only two real-life studies have been published on this issue, one with only 51 patients [10] and a recent study [21] with a smaller sample size (*n* = 198) than in the present investigation (*n* = 249).

The total crude infection-related mortality in these patients was 11.2%, most frequently due to VAP (40%), followed by pneumonia with COVID-19 co-infection (20%), CAP requiring hospitalization (16.7%), NP (14.5%), bacteremia (11.4%), and skin/soft tissue infections (5.6%). Among the microorganisms, the highest mortality rates were for MRSA (20.8%) and *P. aeruginosa* (16.1%). The mortality rate was <1% in the clinical trials of Cefto-M in the patients with CAP. The difference between the present findings might be explained by their stricter eligibility criteria, with the exclusion of the patients receiving an antibiotic for >24 h in the previous three days and those with aspiration pneumonia, viral respiratory infections, polymicrobial infections, or radiological or clinical suspicions of atypical pneumonia [22]. In the trial for the patients with NP, the total mortality rate was 16.7% and the infection-attribution rate was 5.9%. This major discrepancy with the present findings can again be attributed to the trial eligibility criteria, which excluded the patients receiving systemic antibiotic treatment for >24 h in the previous two days and those with severe kidney failure or liver failure, evidence of infection with ceftazidime- or Cefto-M-resistant pathogens, and clinical circumstances potentially hampering the evaluation of the effectiveness, e.g., sustained shock, active tuberculosis, pulmonary abscess, or post-obstructive pneumonia [23].

Only one patient (0.4%) had a severe complication. However, the treatment was not withdrawn from any patient due to an adverse effect, similar to the findings of a single-center real-life study on the use of Cefto-M in 29 patients with infections in a third-level hospital [24].

Finally, the main factors related to mortality in this cohort of Cefto-M-treated patients were older age (the mean age of the patients was 76.7 years), the presence of sepsis/septic shock, and ICU admission, which have all been independently related to higher infection-related mortality rates in the previous studies [25].

The study was limited by its retrospective design and possible selection bias. Its strengths included its multicenter design, sample size (largest to date), and real-life nature, reflecting as faithfully as possible the utilization of Ceftobiprole-M in routine clinical practices in Spain.

## 4. Materials and Methods

### 4.1. Study Design

This real-life, retrospective, multicenter, observational, and descriptive study on the use of Cefto-M included patients in hospital or receiving OPAT with nosocomial/nosohusial or community-acquired infections from 12 Spanish centers in six autonomous communities (Andalusia, Madrid, Cataluña, Valencia, Murcia, and Cantabria). The study period was from the time of the drug’s approval in 2021 to 31 December 2022. The study was approved by the Provincial Ethics Committee of Granada (ref: 0095-N-22), with no requirement for the informed consent of the patients. All the data were gathered in accordance with the Spanish personal data protection legislation (Organic Law 3/5 December 2018) and the Declaration of Helsinki.

This descriptive study did not involve a pharmacological intervention. The treatments were always prescribed by the attending physicians according to their clinical practice.

The inclusion criteria was as follows: age > 17 years; receipt of Cefto-M as the first-line or rescue treatment for ≥48 h (≥six vials in the patients with normal renal function, creatinine clearance-adjusted in the patients with kidney failure); and ≥30 days of follow-up post-discharge or, in the case of the patients with osteomyelitis o endocarditis, ≥6 months post-discharge.

The exclusion criteria was as follows: pregnancy, allergy to beta-lactams, or any formulation excipient.

### 4.2. Variables and Definitions

The variables of this study included the following: age, sex, ethnicity, days of hospitalization (dates of admission and discharge), prescribing hospital department, age-adjusted Charlson index, and comorbidities.

The infection types in this study included the following: bacteremia (complicated/non-complicated], endocarditis (definite/probable/suspected, native/early prosthetic/late prosthetic/on pacemaker), respiratory infection (upper tract/CAP/NP/VAP), urinary tract infection (UTI), central nervous system infection, spondylodiscitis, osteoarticular infection, intra-abdominal infection, or other foci of infection. The etiology of the infections in this study included the following: community or nosocomial/nosohusial/healthcare-related; sepsis or septic shock, monomicrobial/polymicrobial infection, and co-infection with SARS-CoV-2 (COVID-19).

In this study, Cefto-M was administration as monotherapy or combination therapy (for the same infection); empirical or targeted administration; first-line or rescue (due to poor response to previous antibiotherapy, microbiology results, or toxicity with previous antibiotherapy), and was based on the days of administration, dose, and adverse events.

Previous antibiotic (for same infection) with treatment duration.

The microbiology for this study consisted of the microorganism causing the infection and the antibiogram according to the EUCAST criteria [26]. The EUCAST cutoff points were as follows for: *Staphylococci* (Vancomycin (*S. aureus*): 2; Vancomycin (CoNS): 4; Oxacillin (*S. aureus*): 2; Oxacillin (CoNS): 0.25); *Enterococci* (Vancomycin: 4); *Pneumococci* (Cefepime: 1; Ceftobiprole: 0.5; Vancomycin: 2; Meropenem: 2); *Enterobacteriaceae* (Cefepime: 1; Ceftobiprole: 0.25; Meropenem: 2); and *Pseudomonas aeruginosas* (Cefepime: 0.001; Ceftobiprole: insufficient evidence; Meropenem: 2).

Infection-related mortality at 14 and 28 days (at 6 months for endocarditis or osteomyelitis); readmission for the same reason during the first month; and relapse/recurrence of the infection.

The definitions of the terms used in this study are as follows.

-Nosocomial infection: onset > 72 h after hospitalization.-Nosohusial/nosocomial infection: healthcare-related (day hospital, residence, day center for elderly).-The age-adjusted Charlson comorbidity index was used to estimate the 10-year life expectancy of the patients as a function of their age and the presence of comorbidities at admission for the infectious episode [27].-Sepsis/septic shock: refractory hypotension and end-organ perfusion dysfunction despite adequate fluid resuscitation [28].-Immunodepression: congenital or acquired immunodeficiency or receipt of immunosuppressive treatment [29].-Relapse/recurrence of the infection was defined by a second episode within three months [30].-The adverse effect classification used in this study is as follows.-Mild: required no antidote or treatment; brief hospitalization.-Moderate: required treatment modification (e.g., dose adjustment, combination with another drug) but no interruption of drug administration. A longer hospitalization or prescription of a specific treatment may be needed.-Severe: threatened the life of the patient and mandated an interruption of the drug administration and prescription of a specific treatment.-Lethal: directly or indirectly contributed to the death of a patient.


### 4.3. Sample Size

A sample size of approx. 250 individuals was estimated to be adequate to analyze the use of Cefto-M in routine clinical practices with a confidence interval of 95% and an error of 5%. The information was obtained from the electronic records of the different hospital pharmacy departments, gathering the number of patients to whom the drug was administered based on the type of infection. These data were introduced into an anonymized database in an SPSS format, following the national data protection legislation and the principles of the Declaration of Helsinki.

### 4.4. Statistical Analysis

In a descriptive analysis, the absolute and relative frequencies (%) were calculated for the qualitative variables. The means with standard deviation were calculated for the quantitative variables with a normal distribution and the medians were4 calculated with an interquartile range (IQR) for the variables with a non-normal distribution (Kolmogorov–Smirnov test).

In the bivariate analyses of the mortality-related factors, the chi-squared test was used to compare the qualitative variables, the Student’s *t*-test was used for the quantitative variables a with normal distribution, and the Mann–Whitney U test for those with non-normal distribution. A multivariate logistic regression analysis considered the variables that were statistically significant in a bivariate analysis or deemed relevant (i.e., chronic kidney failure, active hematological or solid organ neoplasia, co-infection by SARS-CoV-2, rescue/first-line treatment).

Ethics approval and consent to participate: This study was approved by the ethics committee of the coordinating center and was exempted from the need to obtain informed consent due to its retrospective design and large size. All the data were gathered in accordance with Spanish personal data protection legislation.

## 5. Conclusions

Ceftobiprole-M is a safe antibiotic, comprising only half of the prescriptions for patients with respiratory infection, that is mainly administered as rescue therapy in pluripathological patients with severe infections. The infection-related mortality was 11.2%, which was largely associated with higher age, the presence of sepsis/septic shock, and ICU admission.

## Figures and Tables

**Figure 1 antibiotics-12-01218-f001:**
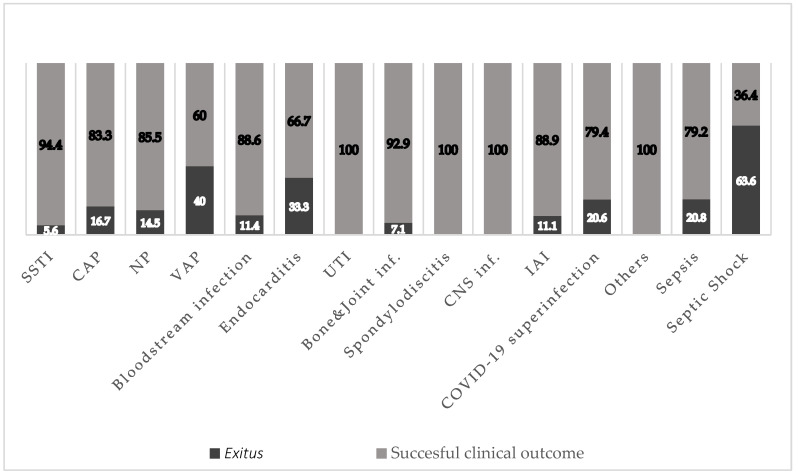
Clinical outcomes by the primary infection type (*n* = 249). SSTI: skin and soft tissue infection, CAP: community-acquired pneumonia; NP: nosocomial pneumonia; VAP: ventilator-associated pneumonia; UTI: urinary tract infection; CNS: central nervous system; IAI: intra-abdominal infection; *Exitus:* death.

**Figure 2 antibiotics-12-01218-f002:**
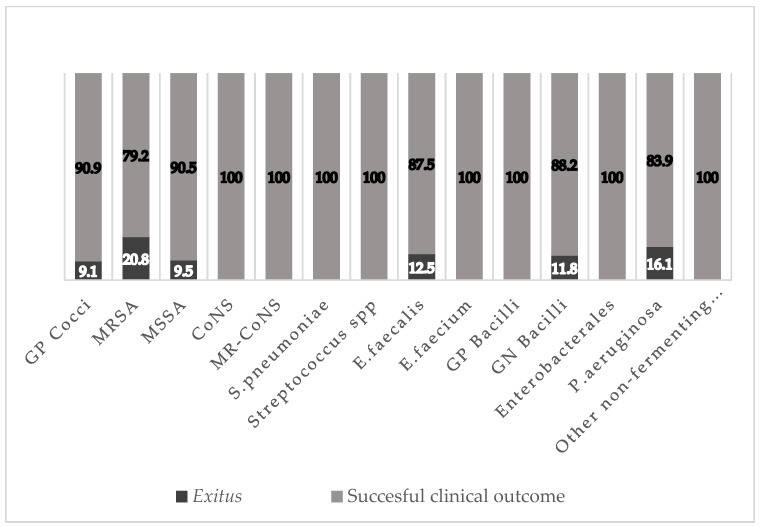
Clinical outcomes of the microbial isolates. GP: gram-positive; MRSA: methicillin-resistant *Staphylococcus aureus*; MSSA: methicillin-susceptible *Staphylococcus aureus*; CoNS: coagulase-negative *Staphylococcus* spp; MR-CoNS: methicillin-resistant coagulase-negative *Staphylococcus* spp; GN: gram-negative; GNB: gram-negative bacilli; *Exitus:* death.

**Table 1 antibiotics-12-01218-t001:** Epidemiological characteristics, comorbidities, and infection pathways.

	Cohort *N* = 249
Age, mean (years), (±SD)	66.6 (±15.4)
Charlson index, median (IQR)	4 (2–6)
Sex, *n* (%)	
Male	148 (59.4)
Female	101 (40.6)
Ethnicity, *n* (%)	
Caucasian	231 (92.8)
Latin	17 (6.8)
African	1 (0.4)
Acquisition of the infection, *n* (%)	
Community-acquired infection	107 (43)
Nosocomial/Nosohusial infection	142 (57)
Presence of sepsis or septic shock, *n* (%)	
Sepsis	66 (26.5)
Septic shock	11 (4.4)
Inpatient departments, *n* (%)	238 (95.6)
Medical department	188 (75.5)
Intensive care unit	12 (4.8)
Surgical department	38 (15.2)
Outpatient antibiotic treatment, *n* (%)	11 (4.4)
Co-infection with SARS-CoV-2 (COVID-19), *n* (%)	34 (13.7)
Comorbidities	
Cardiovascular risk factors, *n* (%)	123 (49.4)
Hypertension	73 (29.3)
Dyslipidemia	11 (4.4)
Obesity	1 (0.4)
≥2 Risk factors	38 (15.2)
Cardiovascular disease, *n* (%)	78 (31.3)
Ischemic heart disease	26 (33.3)
Heart failure	9 (11.5)
Atrial fibrillation/flutter	15 (19.2)
Pacemaker carrier	1 (1.3)
Dilated cardiomyopathy	1 (1.3)
Other conditions	9 (11.5)
≥2 Conditions	17 (21.8)
Respiratory diseases, *n* (%)	74 (29.7)
Chronic obstructive pulmonary disease (COPD)	29 (39.2)
Obstructive sleep apnea (OSA)	9 (12.2)
Thromboembolic pulmonary vascular disease (TPVD)	4 (5.4)
Bronchiectasis	8 (10.8)
Asthma	4 (5.4)
Interstitial lung disease	3 (4.1)
Other conditions	6 (8.1)
≥2 Conditions	11 (14.9)
Gastrointestinal and hepatic diseases, *n* (%)	45 (18.1)
Chronic liver disease	18 (40)
Liver cirrhosis	8 (17.8)
Peptic ulcer disease	6 (13.3)
Inflammatory bowel disease	3 (6.7)
Liver transplantation	3 (6.7)
Other conditions	7 (15.6)
Chronic kidney disease, *n* (%)	35 (14.1)
Active solid malignancy, *n* (%)	20 (8)
Active hematologic malignancy, *n* (%)	33 (13.3)
Metabolic disorders, *n* (%)	83 (33.3)
Diabetes mellitus	70 (84.3)
Hypothyroidism	11 (13.3)
Adrenal insufficiency	2 (2.4)
Neurological diseases, *n* (%)	21 (8.4)
Stroke, *n* (%)	14 (5.6)
Psychiatric conditions, *n* (%)	9 (3.6)
Immunocompromised patients, *n* (%)	52 (20.9)
Immunosuppressant drugs therapy, *n* (%)	43 (17.3)
Infection pathway	
Bloodstream infection, *n* (%)	44 (17.7)
Primary bacteremia	37 (14.9)
Catheter-associated bloodstream infection	7 (2.8)
Infective endocarditis, *n* (%)	3 (1.2)
Respiratory tract infections, *n* (%)	139 (55.8)
Nosocomial pneumonia	62 (24.9)
Community-acquired pneumonia	60 (24.1)
Ventilator-associated pneumonia	5 (2)
Soft tissue and skin infection, *n* (%)	54 (21.7)
Diabetic foot infection	20 (37)
Cellulitis	10 (18.5)
Soft tissue abscess	7 (13)
Infected pressure ulcer	7 (13)
Surgical wound infection	6 (11.1)
Myositis	2 (3.7)
Other type	2 (3.7)
Urinary tract infection, *n* (%)	10 (4)
Complicated UTI (pyelonephritis)	5 (50)
Non-complicated UTI	3 (30)
Renal abscess	2 (20)
Central nervous system infection, *n* (%)	8 (3.2)
Ventriculoperitoneal shunt infection	3 (37.5)
Epidural abscess	2 (25)
Cerebral abscess	2 (25)
Meningitis	1 (12.5)
Intra-abdominal infection, *n* (%)	9 (3.6)
Bone and joint infection, *n* (%)	14 (5.6)
Prosthetic joint Infection	6 (42.9)
Osteomyelitis	4 (28.6)
Infectious tenosynovitis	3 (21.4)
Septic arthritis	1 (7.1)
Spondylodiscitis, *n* (%)	3 (1.2)
Other type of infection, *n* (%)	4 (1.6)

**Table 2 antibiotics-12-01218-t002:** Microbial isolates.

	Cohort *N* = 249
General microbial profile, *n* (%)	
No isolation	111 (45)
Positive microbial samples	137 (55)
Microbial profile of isolates, *n* (%)	
Monomicrobial infection	81 (59.2)
Polymicrobial infection	56 (40.8)
Gram-positive cocci, *n* (%)	87 (63.5)
*Staphylococus aureus*	46 (52.9)
MRSA	24 (52.2)
MSSA	21 (45.6)
Non-categorized *Staphylococcus aureus*	1 (2.2)
CoNS	20 (22.9)
*Staphylococcus epidermidis*	15 (75)
*Staphylococcus hemolyticus*	2 (10)
*Staphylococcus hominis*	2 (10)
*Staphylococcus schleiferi*	1 (5)
*Enterococcus* spp.	9 (10.3)
*Enterococcus faecalis*	8 (88.9)
*Enterococcus faecium*	1 (11.1)
*Streptococcus* spp.	10 (11.5)
*Streptococcus pneumoniae*	5 (50)
*Streptococcus anginosus*	4 (40)
*Streptococcus peroris*	1 (10)
Other cocci	2 (2.3)
*Rhottia* spp.	2 (100)
Gram-positive bacilli, *n* (%)	1 (0.7)
*Cutibacterium acnes*	1 (100)
Gram-negative bacilli, *n* (%)	49 (35.8)
Enterobacterales	13 (26.5)
*Klebsiella pneumoniae*	5 (38.5)
*Escherichia coli*	4 (30.8)
*Klebsiella oxytoca*	1 (7.7)
*Proteus mirabilis*	1 (7.7)
*Proteus vulgaris*	1 (7.7)
Non-fermenting gram-negative bacilli	31 (63.2)
*Pseudomonas aeruginosa*	31 (100)
Other gram-negative bacilli	5 (10.2)
*Morganella* spp.	2 (40)
*Hemophilus influenzae*	2 (40)
*Moraxella catarrhalis*	1 (20)

*S. aureus*: *Staphylococcus aureus.* MRSA: methicillin-resistant *Staphylococcus aureus.* MSSA: methicillin-susceptible *Staphylococcus aureus*. CoNS: coagulase-negative *Staphylococcus* spp.

**Table 3 antibiotics-12-01218-t003:** Susceptibility of microbial isolates.

Microorganisms, *n* (%)		Vanco-S	Cloxa-S	Dapto-S	Ceftobi-S	Cefe-S	Mero-S	Pip/Taz-S
*Staphylococcus aureus*	46 (18.4)	35 (97.2)	14 (41.2)	21 (67.7)	6 (100)			
MRSA	24 (9.6)	21 (100)	0 (0)	16 (80)	3 (100)			
MSSA	21 (8.4)	14 (93.3)	14 (100)	5 (45.5)	3 (100)			
*Enterococcus* spp.	10 (4)	5 (100)	NT	0 (0)	1 (100)			
*Streptococcus* spp.	10 (4)	3 (100)	NT	NT	1 (100)			
GNB	49 (20.5)				10 (100)	4 (33.3)	5 (83.3)	16 (84.2)
*Enterobacteriaceae*	13 (5.2)				5 (100)	1 (50)	NT	6 (100)
*Pseudomonas aeruginosa*	31 (12.4)				4 (100)	2 (40)	5 (83.3)	7 (70)
*Hemophilus influenzae*	2 (0.4)				1 (100)	1 (100)	NT	NT

GNB: gram-negative bacilli. Vanco-S: vancomycin-susceptible; Cloxa-S: cloxacillin-susceptible; Dapto-S: daptomycin-susceptible; Ceftobi-S: ceftobiprole-susceptible; Cefe-S: cefepime-susceptible; Mero-S: meropenem-susceptible; Pip/Taz-S: piperacillin-tazobactam-susceptible. NT: not tested.

**Table 4 antibiotics-12-01218-t004:** Outcomes.

	*N* = 249
Total dose of ceftobiprole, median (IQR)	10.5 (7.5–15)
Duration of antibiotic therapy, median (IQR)	7 (5–10)
Treatment regimen, *n* (%)	
Ceftobiprole monotherapy	134 (53.8)
Antibiotic combination	115 (46.2)
Ceftobiprole + Daptomycin	27 (23.5)
Ceftobiprole + Vancomycin	4 (3.5)
Ceftobiprole + Linezolid	8 (7)
Ceftobiprole + Dalbavancin	1 (0.9)
Ceftobiprole + Clindamycin	2 (1.7)
Ceftobiprole + Tigecycline	4 (3.5)
Ceftobiprole + Cloxacillin	3 (2.6)
Ceftobiprole + Ceftazidime	1 (0.9)
Ceftobiprole + Ceftaroline	2 (1.7)
Ceftobiprole + Ceftriaxone	2 (1.7)
Ceftobiprole + Ceftazidime/Avibactam	2 (1.7)
Ceftobiprole + Meropenem	9 (7.8)
Ceftobiprole + Levofloxacin	10 (8.7)
Ceftobiprole + Ciprofloxacin	4 (3.5)
Ceftobiprole + Piperacillin/Tazobactam	2 (1.7)
Ceftobiprole + Amikacin	6 (5.2)
Ceftobiprole + Azithromycin	10 (8.7)
Ceftobiprole + Metronidazole	13 (11.3)
Ceftobiprole + Trimethoprim/Sulfamethoxazole	7 (6.1)
Ceftobiprole + Doxycycline	2 (1.7)
Ceftobiprole + Fosfomycin	1 (0.9)
Ceftobiprole + Antifungal agents	6 (5.2)
Ceftobiprole + Antiviral agents	2 (1.7)
Length of hospital stay, median (IQR)	20 (13–32)
Ceftobiprole as empirical treatment, *n* (%)	169 (67.9)
Appropriate empirical treatment, *n* (%)	140 (82.8)
Prescription of Ceftobiprole, *n* (%)	
As first-line treatment	74 (29.7)
As second-line or more	175 (70.3)
Reason for switching to Ceftobiprole, *n* (%)	
Failure of previous antibiotic treatment	84 (48)
Toxicity/adverse effects of previous antibiotic treatment	3 (1.7)
Guided by microbiological results	65 (37.1)
Other reasons (or combination of previous)	23 (13.1)
Recurrence and readmission, *n* (%)	
Recurrence of infection (in the first month)	3 (1.2)
Hospital readmission	15 (6)
Mortality, *n* (%)	
Total mortality	54 (21.7)
Non-related-to-infection mortality	26 (10.4)
Related-to-infection mortality	28 (11.2)
14-day mortality	17 (60.7)
28-day mortality	9 (32.1)
6-month mortality	2 (7.1)

**Table 5 antibiotics-12-01218-t005:** Adverse drug effects.

	*N* = 249
Total adverse effects, *n* (%)	9 (3.6)
Severity of adverse effects, *n* (%)	
Mild	4 (1.6)
Moderate	4 (1.6)
Severe	1 (0.4)
Adverse effects by symptoms, *n* (%)	
Elevated liver enzymes	3 (1.2)
Gastrointestinal symptoms	2 (0.8)
Urticaria-like cutaneous rash	1 (0.4)

**Table 6 antibiotics-12-01218-t006:** Mortality risk factors: bivariate and multivariate analyses.

	Non-Survivor *N* = 31	Survivor *N* = 219	Bivariate *p* *	Multivariate HR, 95% IC
Age (±DS)	76.7 (±13.3)	65.3 (±15.2)	0.0001	1.1 (1.04–1.16)
Charlson index, mean (IQR)	4.5 (4–6.75)	4 (2–6)	0.253	
Sex, *n* (%)				
Men	20 (71.4)	128 (57.9)	0.17	
Women	8 (28.6)	93 (44.1)		
Ethnicity, *n* (%)				
Caucasian	27 (96.4)	204 (92.3)		
Latin	1 (3.6)	16 (7.2)	0.718	
African	0 (0)	1 (0.5)		
Inpatient department, *n* (%)	26 (83.9)	212 (95.9)	0.9	
Medical services	24 (92.3)	167 (78.8)		
Infectious diseases	2 (7.1)	55 (24.9)	0.035	0.19 (0.03–1.2)
Internal medicine	9 (32.1)	43 (19.5)	0.12	
Pneumology	2 (7.1)	37 (16.7)	0.27	
Intensive care unit	8 (28.6)	4 (1.8)	0.001	42.02 (4.49–393.4)
Hematology	1 (3.6)	10 (4.5)	0.25	
Oncology	2 (7.1)	14 (6.3)	0.27	
Surgical services	2 (7.1)	36 (16.3)	0.27	
OPAT, *n* (%)	2 (7.1)	9 (4.1)	0.36	
Comorbidities, *n* (%)				
Cardiovascular risk factors	22 (78.6)	101 (45.7)	0.001	1.67 (0.49–5.62)
Cardiovascular disease	6 (21.4)	72 (32.6)	0.231	
Pulmonary disease	10 (35.7)	64 (29)	0.461	
Gastrointestinal and hepatic disease	5 (17.9)	40 (18.1)	0.975	
Chronic kidney disease	4 (14.3)	31 (14)	0.97	0.94 (0.21–4.33)
Active solid malignancy	3 (10.7)	17 (7.7)	0.526	1.81 (0.289–11.41)
Hematological malignancy	4 (14.3)	29 (13.1)	0.864	1.21 (0.24–6.16)
Metabolic disorders	11 (39.3)	72 (32.6)	0.478	
Neurological diseases	6 (21.4)	15 (6.8)	0.019	2.59 (0.69–9.85)
Psychiatric disorders	0 (0)	9 (4.1)	0.6	
Stroke	3 (10.7)	11 (5)	0.199	
Immunosuppression	10 (35.7)	42 (19)	0.04	2.03 (0.52–7.88)
COVID-19 superinfection, *n* (%)	7 (25)	27 (12.2)	0.063	2.08 (0.43–10.12)
Number of pathway infection, mean (IQR)	1 (1–1)	1 (1–1)	0.945	
Pathway infection, *n* (%)				
Bloodstream infection	5 (17.9)	39 (17.6)	0.978	
Infective endocarditis	1 (3.6)	2 (0.9)	0.223	
Communitary-acquired pneumonia	10 (35.7)	50 (22.6)	0.127	
Nosocomial pneumonia	9 (32.1)	53 (24)	0.347	
Ventilator-associated pneumonia	2 (7.1)	3 (1.4)	0.04	0.12 (0.004–3.89)
Skin and soft tissue infection	3 (10.7)	51 (23.1)	0.135	
Urinary tract infection	0 (0)	10 (4.5)	0.251	
Central nervous system infection	0 (0)	8 (3.6)	0.306	
Intra-abdominal infection	1 (3.6)	8 (3.6)	0.99	
Bone and joint infection	1 (3.6)	13 (5.9)	0.617	
Spondylodiscitis	0 (0)	3 (1.4)	0.535	
Other type of infection	0 (0)	4 (1.8)	0.473	
Sepsis or shock	16 (57.1)	61 (27.6)	0.0001	2.94 (1.01–8.54)
Microbiology and acquisition of the infection, *n* (%)				
Microbial isolation			0.758	
Monomicrobial infection	9 (32.1)	84 (38)		
Polymicrobial infection	6 (21.4)	50 (22.6)		
Place of acquisition of the infection			0.762	
Communitary-acquired infection	12 (42.9)	95 (43)		
Nosocomial infection	10 (35.7)	90 (40.7)		
Nosohusial infection	6 (21.4)	36 (16.4)		
GPC	8 (28.6)	80 (36.2)	0.426	
MRSA	5 (17.9)	19 (8.6)	0.118	
MSSA	2 (7.1)	19 (8.6)	0.794	
CoNS	0 (0)	20 (9)	0.097	
*Enterococcus faecalis*	1 (3.6)	7 (3.2)	0.909	
*Streptococcus pneumoniae*	0 (0)	5 (2.3)	0.421	
GNB	6 (21.4)	45 (20.4)	0.895	
*Pseudomonas aeruginosa*	5 (17.9)	26 (11.8)	0.358	
Antimicrobial therapy				
Total dose of ceftobiprole (mg), mean (IQR)	9 (4.5–12.75)	10.5 (7.5–15)	0.049	0.91 (0.73–1.12)
Length of ceftobiprole therapy (days), mean (IQR)	6 (3–8.5)	7 (5–10)	0.029	1.08 (0.82–1.4)
Therapy regimen:				
Ceftobiprole monotherapy, *n* (%)	16 (57.1)	118 (53.4)	0.708	
Antibiotic combination, *n* (%)	12 (42.9)	103 (46.6)		
Prescription of ceftobiprole:				
First-line, *n* (%)	6 (21.4)	68 (30.8)	0.308	1.34 (0.4–4.49)
Rescue therapy, *n* (%)	22 (78.6)	153 (69.2)		
Empirical treatment, *n* (%)	22 (78.6)	146 (66.1)	0.183	

OPAT: outpatient parenteral antibiotic therapy; GPC: gram-positive cocci; CoNS: coagulase-negative staphylococcus; GNB: gram-negative bacilli; MRSA: methicillin-resistant *Staphylococcus aureus;* MSSA: methicillin-susceptible *S. aureus*. HR: hazard ratio, 95% CI: 95% confidence interval. * *p* < 0.05 as significant.

## Data Availability

The researchers confirm the accuracy and availability of the data used in this study.

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
