# Peer review of "Cefto Real-Life Study: Real-World Data on the Use of Ceftobiprole in a Multicenter Spanish Cohort"

_antibiotics, 2023, doi:10.3390/antibiotics12071218_

Round 1

Reviewer 1 Report

Abstract:

The abstract is concise and well written

Introduction:

 It is unclear to the reader what is driving “off label” ceftobiprole medocaril use in this setting (e.g. multi-drug resistant bacteria, treatment failures, local guidelines, economics). If this is known it would be helpful to the reader to explain this.

Reference is made to the merino study (Line 53) but it should be remembered that this was an RCT on blood-stream infections and discrepant Piperacillin-tazobactam susceptibility discrepancies may have driven findings. Albeit a real world limitation of the use of this antibiotic. A better argument for the use of Cetobiprole medocaril in my opinion would be the sparing of carbapenems in the era of Carbapenemase-producing enterobacterales (CPE) and the argument that the authors do make regarding retained activity against poring/efflux pump expressing Pseudomonas aeruginosa.

A further argument as to the need for this study should note that previous real world studies have included very small numbers (with appropriate referencing).

Results:

Reference to table should be consistent. Line 83 “(also see table 1), Line 90 (also see Table 1).  I believe the correct format would be (Table 1) as you have used for Table 2 later in manuscript. Suggest capitalize consistently. There is no full stop at the end of Line 90.

Table 1 need to be re-formatted, the tabulated lines makes it difficult for the reader to compare lines of data.

The antimicrobial susceptibility testing methods used needs to be described. This is particularly important when describing rates of Piperacillin-tazobactam susceptibility (see comment above re Merino). Presumably as this is multi-center study a variety of commercial methods were used.  Table 2 is better formatted and easy for the reader to follow although the Eucast cutoff points are out of line. 

(also see Figure 1) should just read (Figure 1) Line 122.

Table 4 and Table 5 are not easy to read with current format/ indentation/ bullet points. Clearer formatting would assist the reader.

Figure 1 and 2 : Please define “Exitus” for the reader, is this mortality or clinical failure? These figures are easy to read, notably the font size and type is inconsistent.

Discussion:

Because one of your major conclusions is that half of all prescriptions of Ceftobiprole medocaril is for non-approved/ off-label use it would be good to have a discussion at least hypothesizing why your clinicians are using it this way. Its use in GNB infections does not seem to be driven by carbapenem resistance rates (notable that 83.3% were susceptible).  Notably all the MRSA infections were Vancomycin susceptible. Given the median length of prescription was 7 days presumably this is not all empirical therapy.

It would assist the reader if you were to offer your local expertise and commentary on why you believe this antibiotic is being used in this way.

Minor punctuation/ consistency recommendations included above.

Author Response

RESPONSES TO REVIEWER 1

Abstract:

The abstract is concise and well written

Response: We are grateful for this positive evaluation.

Reviewer:

Introduction:

 It is unclear to the reader what is driving “off label” ceftobiprole medocaril use in this setting (e.g., multi-drug resistant bacteria, treatment failures, local guidelines, economics). If this is known it would be helpful to the reader to explain this. Reference is made to the merino study (Line 53) but it should be remembered that this was an RCT on blood-stream infections and discrepant Piperacillin-tazobactam susceptibility discrepancies may have driven findings. Albeit a real-world limitation of the use of this antibiotic. A better argument for the use of Ceftobiprole medocaril in my opinion would be the sparing of carbapenems in the era of Carbapenemase-producing enterobacterales (CPE) and the argument that the Responses do make regarding retained activity against poring/efflux pump expressing Pseudomonas aeruginosa. A further argument as to the need for this study should note that previous real-world studies have included very small numbers (with appropriate referencing).

Response: We have clarified the reasons for our off-label use of ceftobiprole in the revised text

We now report that the Merino study was an RCT and that the results did not support the use of Piperacillin-tazobactam in the bloodstream infections under study.

We have also incorporated the reviewer´s suggestion on the sparing of carbapenems in relation to carbapenemase-producing Enterobacteriaceae.

We have also noted the limited sample sizes in previous real-life studies, as recommended.

Reviewer:

Results:

Reference to table should be consistent. Line 83 “(also see table 1), Line 90 (also see Table 1).  I believe the correct format would be (Table 1) as you have used for Table 2 later in manuscript. Suggest capitalize consistently. There is no full stop at the end of Line 90. 

Response: We are grateful to the reviewer for detecting these inconsistencies. All of these changes have been made.

Table 1 need to be re-formatted, the tabulated lines makes it difficult for the reader to compare lines of data.

Response: The table has been re-formatted accordingly.

The antimicrobial susceptibility testing methods used needs to be described. This is particularly important when describing rates of Piperacillin-tazobactam susceptibility (see comment above re Merino). Presumably as this is multi-center study a variety of commercial methods were used.  Table 2 is better formatted and easy for the reader to follow although the EUCAST cutoff points are out of line.  

Response: All microbiology laboratories in Spain implement EUCAST criteria to determine the susceptibility of microorganisms to antimicrobials. The testing methods in the laboratories participating in our study are those commonly used in Spanish clinical practice.

Reviewer: (also see Figure 1) should just read (Figure 1) Line 122.

Response: This change has been made.

Table 4 and Table 5 are not easy to read with current format/ indentation/ bullet points. Clearer formatting would assist the reader.

Response: These tables have been re-formatted accordingly.  

Figure 1 and 2: Please define “Exitus” for the reader, is this mortality or clinical failure? These figures are easy to read, notably the font size and type is inconsistent.

Response: Exitus (Latin term used in Spain) = death, as now defined in the footnotes. The fonts and font sizes are now consistent.

Discussion:

Because one of your major conclusions is that half of all prescriptions of Ceftobiprole medocaril is for non-approved/ off-label use it would be good to have a discussion at least hypothesizing why your clinicians are using it this way. Its use in GNB infections does not seem to be driven by carbapenem resistance rates (notable that 83.3% were susceptible).  Notably all the MRSA infections were Vancomycin susceptible. Given the median length of prescription was 7 days presumably this is not all empirical therapy. It would assist the reader if you were to offer your local expertise and commentary on why you believe this antibiotic is being used in this way.

Response: We have added the following text in the Discussion:  

In addition, Cefto-M was used to treat Gram-negative bacterial (GNB) infections to avoid the utilization of carbapenems and help reduce the incidence of carbapenemase-producing Enterobacteriaceae. Furthermore, in cases of infection by methicillin-resistant CGP such as MRSA, which were all susceptible to vancomycin, Cefto-M was prescribed instead of this lipoglycopeptide due to its rapid bactericidal activity, high volume of distribution to tissues, and excellent safety profile.

Comments on the Quality of English Language

Minor punctuation/ consistency recommendations included above

Response: These recommendations were all followed.

We are very grateful to the reviewer for these helpful comments and suggestions.

Reviewer 2 Report

This manuscript is well information for real-life study in Spain about the routine administration of Cefto-M to patients in the multicenter. It is a fair report that might be published in the Antibiotics journal with major correction and responses as below;

1.      There are several typo errors. Some will be mentioned, but not all. Please double check throughout the manuscript.

2.      Writing scientific names of microorganisms should be followed e.g. italics, full name at the first mention (e.g. Line 53, E. coli or Klebsiella pneumoniae; Line 97, Enterococcus spp.). Please double check throughout the manuscript. 

3.      In the Abstract part, please check whether writing structure is fit with the journal.

4.      In the introduction part, please provide the references for each specific statement.

5.      In the result part,

a.      Lines 93 please mention the total number of the isolates used for calculation. This makes reader a bit confuse.

b.      Seems not mention about 20 of CoNS in Table 2.

c.       No information for Morganella spp. and Moraxella spp. in Table 2. Why not? These microorganisms and their AST results should be included in Table 2.

d.     Line 102, no supplementary data is available to be reviewed.

e.      Table 1. It seems that some data is not valid enough e.g. Sum of numbers of inpatient department (200+12+38) and outpatient antibiotic treatment (11) whether should be exactly at 249? So please double your data before resubmit this again.

f.        Table 2. It seems not valid enough, as some information does not match with the in-text information (e.g., GNB, Enterococcus spp.).

g.      EUCAST cutoff points should be moved to Materials and Methods part.

h.      Table 4 is not valid enough, as Multivariate OR values are quite shifted. This makes readers difficult to evaluate the information. So please double your data before resubmit this again.

6.      In the discussion part, Lines 207-210, I agree with the statement but not sure whether this reference mentions it?

7.      Please recheck references in MDPI style

Author Response

RESPONSES TO REVIEWER 2

This manuscript is well information for real-life study in Spain about the routine administration of Cefto-M to patients in the multicenter. It is a fair report that might be published in the Antibiotics journal with major correction and responses as below;

  1. There are several typo errors. Some will be mentioned, but not all. Please double check throughout the manuscript.

Response: The manuscript has been thoroughly checked to amend typographic errors.

  1. Writing scientific names of microorganisms should be followed e.g. italics, full name at the first mention (e.g. Line 53, E.coli or Klebsiella pneumoniae; Line 97, Enterococcus spp.). Please double check throughout the manuscript.  

Response: These corrections have been made.

  1. In the Abstract part, please check whether writing structure is fit with the journal.

Response: This has been done

  1. In the introduction part, please provide the references for each specific statement. 

Response: This has been done (citations highlighted in yellow).

  1. In the result part, 
  2. Lines 93 please mention the total number of the isolates used for calculation. This makes reader a bit confuse. 

Response: These data are given in lines 98-99: Microbiological isolates were obtained from 137 patients (55 %) and were polymicrobial in 56 (40.6%).

  1. Seems not mention about 20 of CoNS in Table 2.
  2. No information for Morganellaspp. and Moraxella spp. in Table 2. Why not? These microorganisms and their AST results should be included in Table 2.

Responses: In table 2, we exhibit all microorganisms tested for susceptibility to ceftobiprole in the participating hospitals for comparison with other antibiotics. CoNS, Moraxella, and Morganella were not included because data were available on their susceptibility to other antibiotics but not to ceftobiprole.

  1. Line 102, no supplementary data is available to be reviewed.

Response: This error has been remedied (now reported after Table 1).

  1. Table 1. It seems that some data is not valid enough e.g. Sum of numbers of inpatient department (200+12+38) and outpatient antibiotic treatment (11) whether should be exactly at 249? So please double your data before resubmit this again. 

Response: This was poorly expressed in the table, where ICU was considered under the heading of Medical Department, and this was not made clear. The numbers for ICU are now presented separately from those for Medical Department.

  1. Table 2. It seems not valid enough, as some information does not match with the in-text information (e.g., GNB, Enterococcus spp.).

Response: This was an unfortunate error in the text, which has now been corrected.

  1. EUCAST cutoff points should be moved to Materials and Methods part.

Response: This has been done (lines 294-298).

  1. Table 4 is not valid enough, as Multivariate OR values are quite shifted. This makes readers difficult to evaluate the information. So please double your data before resubmit this again.

Response: Table 4 has been re-formatted accordingly.

  1. In the discussion part, Lines 207-210, I agree with the statement but not sure whether this reference mentions it?

Response: We can confirm that this statement is supported by the reference: Kumar A, Roberts D, Wood KE, Light B, Parrillo JE, Sharma S, Suppes R, Feinstein D, Zanotti S, Taiberg L, Gurka D, Kumar A, Cheang M. Duration of hypotension before initiation of effective antimicrobial therapy is the critical determinant of survival in human septic shock. Crit Care Med. 2006; 34:1589-96.

  1. Please recheck references in MDPI style

Response: This has been done.

We are grateful to the reviewer for these comments and suggestions, which have helped to strengthen our paper.

Round 2

Reviewer 2 Report

The manuscript is needed to minor correction as the attached file. Some typo error and scientific writing still exists in the revised manuscript. Please correct it and make it suitable for publication.

Author Response

Thank you very much for your comments, we have made the corrections of some typo error and scientific writing, suggested by the reviewer. We hav made our paper suitable for publication. 
